

# Crucial biological functions of CCL7 in cancer

Yangyang Liu[1,*], Yadi Cai[1,*], Li Liu[2], Yudong Wu[3] and Xiangyang Xiong[2]

[1] First Clinical Medical College, School of Medicine, Nanchang University, Nanchang, People's Republic of China
[2] Department of Biochemistry and Molecular Biology, School of Basic Medical Sciences, Nanchang University, Nanchang, People's Republic of China
[3] Department of Breast Surgery, Jiangxi Provincial Cancer Hospital, Nanchang, People's Republic of China
* These authors contributed equally to this work.

## ABSTRACT

Chemokine (C-C motif) ligand 7 (CCL7), a CC chemokine, is a chemotactic factor and attractant for various kinds of leukocytes, including monocytes and neutrophils. CCL7 is widely expressed in multiple cell types and can participate in anti-inflammatory responses through binding to its receptors to mediate the recruitment of immune cells. Abnormal CCL7 expression is associated with certain immune diseases. Furthermore, CCL7 plays a pivotal role in tumorigenesis. CCL7 promotes tumor progression by supporting the formation of the tumor microenvironment and facilitating tumor invasion and metastasis, although some studies have suggested that CCL7 has tumor suppressor effects. In this review, we summarize the currently available information regarding the influence of CCL7 on tumors.

## INTRODUCTION

Chemokines are a class of cytokines that control cell-directed migration. Inflammatory cytokines, growth factors and exogenous stimuli can induce the production of chemokines, which then selectively attract leukocytes to sites of tissue damage and infection, and these leukocytes mediate the immune responses (*Balkwill, 2004*). Chemokines are small (8–10 kDa), basic, heparin-binding proteins that have 20–70 percent homology at the amino acid sequence level (*Balkwill, 2004*; *Van Coillie, Van Damme & Opdenakker, 1999*). Based on the number and array of conserved cysteines, chemokines are classified into four groups: Cys-X-Cys (CXC), Cys-Cys (CC), Cys (C), and Cys-X3-Cys (CX3C) (*Christopherson & Hromas, 2001*). CXC and CC chemokines, the first have been discovered, contain four highly conserved cysteines, of which the first two are separated by one amino acid (CXC) or are adjacent (CC). Members of the C family possess only two conserved cysteines, the second and fourth conserved cysteine residues, and three amino acids are inserted between the first two cysteine residues in the CX3C family (*Kondo et al., 2000*). Chemokines depend on binding to their receptors expressed on various responsive cells to exert function. Currently, more than

Corresponding author
Xiangyang Xiong,
xiangyangxiong@ncu.edu.cn

50 chemokines and at least 20 corresponding receptors have been identified (*Palomino & Marti, 2015*).

Chemokine (C-C motif) ligand 7 (CCL7), also known as monocyte chemotactic protein 3 (MCP-3), is a member of the CC subfamily that was first characterized from osteosarcoma supernatant (*Van Damme et al., 1992*). CCL7 is expressed in various types of cells under physiological conditions, including in stromal cells, airway smooth muscle cells, and keratinocytes, and in tumor cells under pathological conditions. CCL7 is a potent chemoattractant for a variety of leukocytes, including monocytes, eosinophils, basophils, dendritic cells (DCs), NK cells and activated T lymphocytes. As a chemotactic factor, CCL7 recruits a leukocyte subtype to infected tissues to address pathologic invasion and fine-tune the immune response. However, abnormal increase of CCL7 exacerbates the deterioration of various disorders, like lesional psoriasis (*Brunner et al., 2015*), acquired immunodeficiency syndrome (*Atluri et al., 2016*), acute neutrophilic lung inflammation and pulmonary fibrosis (*Choi et al., 2004*; *Mercer et al., 2014*). More recently, the autocrine and paracrine roles of CCL7 in cancer progression have received increasing attention, although the potential molecular cues involved are incompletely understood. In this review, we provide an overview of CCL7 in tumorigenesis, and the data presented herein are expected to provide new approaches for tumor therapy.

## SURVEY METHODOLOGY

PubMed database was used for related literature search using the keyword "CCL7," "cancer," "MCP-3" and "tumorigenesis."

## THE STRUCTURE AND REGULATION OF CCL7

The human CCL7 gene is located on chromosome 17q11.2-12 (*Van Coillie, Van Damme & Opdenakker, 1999*). This region harbors the gene for the MCP subset of CC chemokines and can be distinguished from the syntenic MIP-1α locus. SCYA7 is the locus symbol of the CCL7 gene, and there is a double microsatellite $(CA)_n$-$(GA)_n$ at the 5′-end of this gene (*Opdenakker et al., 1994*). As shown in Fig. 1, the gene is composed of three exons and two introns. The first exon contains a 5′-untranslated region (5′-UTR), an N-terminal signal sequence (23 amino acids), and the first two amino acids of the mature protein. The second exon contains amino acids 3–42 of the mature protein. The final exon encodes the C-terminal region of the protein, a 3′-UTR containing one or more destabilizing AU-rich sequences and a polyadenylation signal in eukaryotes (*Van Coillie, Van Damme & Opdenakker, 1999*). The transcriptional start site is located 16 bp downstream of the promoter TATA box. There are some balanced transcriptional elements upstream of the promoter. CRE element and Ets-like element strongly diminish promoter activity. There are two enhancers, one is located at position ranging from −172 to −110 and the other is located at position −37 and only 21 bp upstream from the promoter TATA box, which is named AP-1 like element (*Murakami et al., 1997*).

Full-length CCL7 is 99 amino acids after transcription and translation and contains a 23-amino acid signal peptide. The mature protein of 76 amino acids is secreted after

 

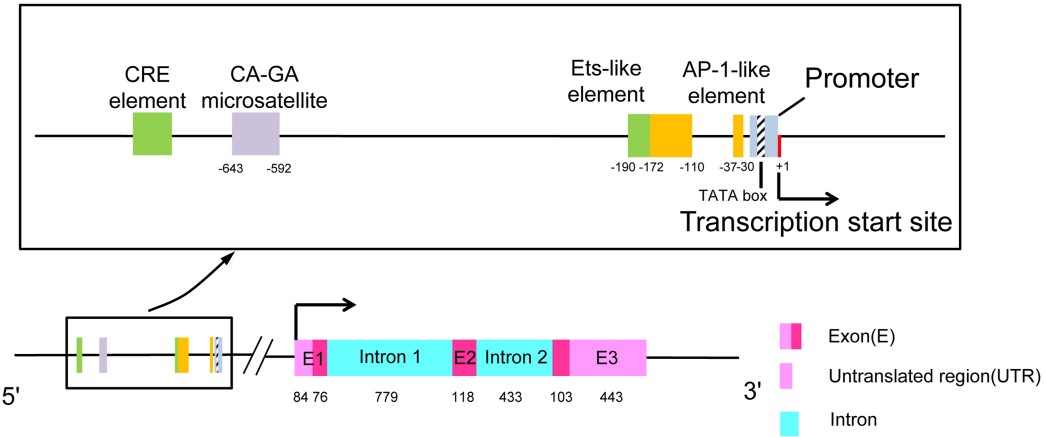

**Figure 1 The structure of chemokine CCL7.** The rectangular box shows the upstream transcriptional regulatory elements of the promoter in detail. The length of exons and introns is expressed in base pairs.

cleavage of the signal peptide. CCL7 exists in a general monomeric form, differing from the dimer formed in a highly concentrated solution (*Ali et al., 2005*). There are four different glycotypes (11, 13, 17, and 18 kDa) of CCL7 formed by N- and O-glycosylation in COS cells (*Van Coillie, Van Damme & Opdenakker, 1999*). Glycosaminoglycans (GAGs) on the surface of endothelial cells interact with amino acids 44, 46, and 49 of CCL7 to produce a complex that is capable of activating transendothelial leukocyte migration *in vitro* (*Ali et al., 2005*). Mutation of these sites results in a lack of affinity for GAGs and the inhibition of CCL7-mediated transendothelial leukocyte migration; however, normal receptor affinity is preserved, and normal intracellular $Ca^{2+}$ flux can be induced in mononuclear leukocytes. N-terminal amino acid addition or deletion or CCL7 sequence truncation at other sites by matrix metalloproteinase 2 (MMP2) forces CCL7 to function as a receptor antagonist, inhibiting the activity of intact CCL7 *in vivo* (*Masure et al., 1995*; *McQuibban et al., 2000*).

To guard the body against superfluous damage, restrictive immune reactions in infected locations play an important role in strictly supervising chemokine production. An overview of the cellular sources and expression regulation is given in Table 1. Unfortunately, the specific response elements and signaling pathways involved are not very clear. Studies on the role of latent signaling pathways in regulating CCL7 through certain cytokines (e.g., IL-1β and IFN-β) should be performed.

## CCL7 RECEPTORS

Chemokines often have shared and exclusive chemokine receptors. The genes encoding chemokine-binding receptors are localized on chromosome 3p21-22, and the receptors can be classified as a subtype of G protein-coupled seven-transmembrane receptors (*Griffith, Sokol & Luster, 2014*). CC chemokine receptor (CCR) 1, CCR2, and CCR3 are widely acknowledged as the main functional receptors of CCL7 (*Palomino & Marti, 2015*). It has also been reported that CCL7 can bind to CCR5 and CCR10 (*Jung et al., 2010*;

**Table 1 The regulation of CCL7.**

| Chemokine | Producer cell | Inducer(+)/inhibitor(−) | Reference |
|---|---|---|---|
| | Endothelial cells | + PMA | *Kondo et al. (2000)* |
| | | − forskolin | |
| | Mononuclear cells | + LAM, BCG, PHA, TNF-α, LPS, IL-1β, | *Menten et al. (1999)*, |
| | | + IFN-γ, IFN–α, IFN-β, measles virus | *Vouret-Craviari et al. (1997)* |
| | | − IL-10, IL-13 | |
| CCL7 | Fibroblasts cells | + IL-1β, IFN-γ | *Menten et al. (1999)* |
| | Human airway smooth muscle cells (HASMCs) | + IL-1β | *Pype et al. (1999)* |
| | Vascular smooth muscle cells (VSMCs) | + TNF-α | *Zhao et al. (2013)* |
| | Renal cell carcinoma (RCC) cells | − Let-7d | *Su et al. (2014)* |
| | Osteosarcoma cells (MG-63) | + IL-1β, PMA, IFN-β, measles virus | *Murakami et al. (1997)* |
| | Myelomonocytic cells (THP-1) | + PMA, LPS | *Murakami et al. (1997)* |
| | Lymphoma cells (U937) | + PMA | *Murakami et al. (1997)* |

**Note:**
PMA, phorbol12-myristate13-acetate; LAM, mycobacterial lipoarabinomannan; PHA, phytohaemagglutinin.

*Lee et al., 2016*; *Van Coillie, Van Damme & Opdenakker, 1999*). Chemokine receptors are approximately 339–373 amino acids long, encompassing three intracellular loops, three extracellular loops, and the free N- and C-termini. Compared to other kinds of G protein-coupled receptors, chemokine receptors are small, containing a short N-terminal region and a very short third intracellular loop (*Murphy, 1996*). Serine and threonine residues are abundant in the C-terminal region, serve as phosphorylation sites after receptor activation, and are responsible for receptor desensitization. In addition, the extracellular loops and N-terminus are required for ligand binding (*Van Coillie, Van Damme & Opdenakker, 1999*). The expression profiles of chemokine receptors are complex and individualized, and these profiles can be affected by cell lineage, differentiation state, and microenvironmental factors such as chemokine concentration, the presence of inflammatory cytokines and hypoxia (*Balkwill, 2004*; *Palomino & Marti, 2015*). A shared characteristic of these receptors is that they showed restricted expression in leukocyte subtype but can be expressed in some tumor types (*Nelson & Krensky, 2001*). CCR1 can be expressed on the surface of breast cancer cells, accelerating tumor angiogenesis in the tumor microenvironment (*Rajaram et al., 2013*). Exogenous TNF-α can provide support for the recruitment of bone marrow-derived mesenchymal stromal cells (BM-MSCs) into the tumor microenvironment, wherein these cells are differentiate into tumor-resident MSCs that overexpress CCR2 ligands. Similarly, tumor infiltrating CCR2-positive monocytes/macrophages have a strong tumor-promoting effect (*Arendt et al., 2013*; *Ren et al., 2012*). Thus, tumor-resident MSCs drive tumorigenesis via the chemokine-CCR2-positive monocyte/macrophage axis (*Ren et al., 2012*). The targeted treatment of MSC-based cancer may benefit from knockdown of *Ccr2* (*Mueller et al., 2011*; *Ren et al., 2008*; *Sarkar et al., 2010*). CCR3 is expressed in prostate cancer cells, and its upregulated expression has been shown to correlate significantly with cancer cell migration and invasion (*Laurent et al., 2016*).

Because of overlap in the structures of ligands and receptors, some chemokines bind to multiple receptors, and receptors can share multiple chemokines from the same general family. Thus, the network of CCL7 and its receptors is complex. CCL7 shares receptors not only with CCL2 on monocytes and basophils but also with RANTES on basophils and eosinophils (*Dahinden et al., 1994*; *Noso et al., 1994*; *Sozzani et al., 1994*), as well as with MIP-1α on basophils, eosinophils, and neutrophils. CCL7 may also affect additional leukocyte receptors and interconnected signaling pathways to exert its function, and blocking CCL7 binding to receptors may represent a new therapeutic strategy (*Ben-Baruch et al., 1995*).

## THE PHYSIOLOGICAL FUNCTION OF CCL7

CCL7 appears to influence leukocyte migration, including spreading, diapedesis, and extravasation (*Weber et al., 1999*), and subsequent events associated with inflammation-related immune responses. Exogenous or endogenous signals trigger a cascade, and then, CCL7 selectively recruits leukocytes that express associated receptors to migrate along the concentration gradient to sites of inflammation. In monocyte mobilization from BM to blood circulation, the positive effect of CCL7 is especially prominent, and a similarly strong effect is also observed in the recruitment of monocytes to sites of inflammation (*Tsou et al., 2007*). A previous study reported that CCL7, as the only member of the CC subfamily, can induce steady neutrophil migration by increasing intracellular $Ca^{2+}$ flux; this function is similar to that of members of the CXC chemokine family (*Fioretti et al., 1998*). These data provide a basis for placing CCL7 in an absolutely dominant position in inflammatory reactions (*Xu et al., 1995*). In addition, the speed of immune responses is dissimilar in different cells. Upon stimulation by proinflammatory cytokines such as IL-1β and TNF-α, the response is immediate, and CCL7 is expressed by fibroblasts, epithelial cells, and endothelial cells. Correspondingly, there is a "prolonged effect" in T lymphocytes, which initiate expression 3–5 days after being activated. The timing and locations of immune responses are amplified because of these late expression dynamics (*Song, Nikolcheva & Krensky, 2000*).

CCL7 greatly impacts diverse immune responses, involving antiviral, anti-bacterial, and antifungal immunity. Mice that are genetically deficient in *Ccl7* (*Ccl7$^{-/-}$* mice) have a markedly increased viral burden in the central nervous system and increased mortality, accompanied by a profound decrease in monocyte and neutrophil quantity, when infected by West Nile virus (*Bardina et al., 2015*). CCL7 can also facilitate the elimination of *Listeria monocytogenes* infection by increasing the recruitment of inflammatory monocytes and TNF/iNOS-producing dendritic cells (*Serbina, Shi & Pamer, 2012*). Additionally, interplay between Toll-like receptor 9 (TLR9) and the CCL7/CCR2 axis is an important part of protective responses to lung cryptococcal infection. As an important downstream effector of the TLR9 signaling pathway, CCL7 stimulates IFN-γ production and activated CD11b$^+$ DCs accumulation in the early stage of the immune response. During the efferent stage of the immune response, CD4$^+$ and CD8$^+$ T cells, CD11b$^+$ DCs and exudate macrophages accumulate with the help of produced CCL7, and then, CCL7 strongly promotes the clearance of *Cryptococcus neoformans* (*Qiu et al., 2012*). Collectively,

CCL7 is a multipotent chemokine, as evidenced by its involvement in antiviral, anti-bacterial and antifungal immune reactions.

## THE ROLE OF CCL7 IN CANCER

There are two important factors that affect tumor evolution: inherent tumor characteristics and crosstalk between tumor cells and stromal cells in the peripheral tumor microenvironment. As an indispensable component of the tumor microenvironment, stromal cells include fibroblasts, macrophages, adipocytes, and others (*Raffaghello & Dazzi, 2015*). Connections between stromal cells and tumor cells are formed by a variety of soluble factors, including inflammatory cytokines, growth factors and chemokines secreted by stromal cells or tumor cells (*Kojima et al., 2010*; *Mazzocca et al., 2011*; *Sugihara et al., 2015*). CCL7 is an important molecular regulator in the reciprocal interaction between stromal cells and tumor cells, which can not only participate in the tumorigenesis but also exert its antitumor effect in specific contexts. Increased CCL7 levels recruit monocytes to sites at the tumor periphery, which helps in the formation of an environment suitable for carcinoma progression and promotes monocytes to complete phenotypic transformation. In contrast, CCL7 is also able to recruit more leukocytes and activate antitumor immune responses. Here, we will focus on the classification of CCL7 based on its original cell to explain this hypothesis.

### Pro-tumor effects of CCL7

#### Tumor cell-derived

CCL7 can act as a tumor-derived factor that may promote tumor growth, invasion and metastasis in an autocrine manner. CCL7 is upregulated in lung adenomas isolated from old mice, which show marked accumulation of immune cells, and lung adenomas formed on an aging background are more invasive (*Parikh et al., 2018*). CCL7 is expressed at higher levels in metastatic renal cell carcinoma (RCC) than in primary RCC (*Wyler et al., 2014*). High CCL7 expression evokes the recruitment of tumor-associated macrophages (TAMs) that express CCR2 on the surface, resulting in the persistence of increased vascular permeability (*de Vries et al., 2006*; *Medioni et al., 2007*); therefore, tumor cells can cross the blood–brain barrier and move toward brain tissues (*Wyler et al., 2014*). The pro-tumorigenic properties of CCL7 have also been confirmed in colorectal cancer (CRC) cells (*Cho et al., 2012*; *Lee et al., 2016*). CRC cell proliferation, migration and invasion are increased by the overexpression of CCL7 *in vitro* and *in vivo*. In clinical specimens, higher CCL7 expression in liver metastatic tumor tissues suggested that CCL7 promotes CRC liver metastasis (*Cho et al., 2012*). CRC cells stably overexpressing CCL7 by lentiviral transduction show enhanced expression of CCR3 (*Lee et al., 2016*). In addition, via binding to CCR3, CCL7 overexpression activates the ERK/JNK signaling pathway that converges on the downstream pathways of the MAPK cascade, thereby participating in the epithelial-mesenchymal transition (EMT) process that is sufficient to strengthen cancer metastasis capabilities (*Lee et al., 2016*).

Let-7d microRNA is a let-7 family member, and it was originally identified in *Caenorhabditis elegans*. Let-7d is expressed in a time-specific manner, with constitutively

high expression in distinct adult tissues (*Thomson et al., 2004*). As a tumor suppressor (*Boyerinas et al., 2010*), let-7d isoforms are frequently downregulated in many human malignancies, such as lung cancer, breast cancer, and hepatocellular carcinoma (HCC) (*Shimizu et al., 2010*; *Yu et al., 2007*; *Zhao et al., 2014*). Let-7d specifically binds to the 3′-UTR of CCL7 mRNA and modulates its expression in a negative feedback manner. The expression of let-7d is downregulated in RCC, in which generous amount of CCL7 is produced (*Su et al., 2014*). Enhanced expression of CCL7 increases macrophage chemotaxis potential, which is proven to promote cancer initiation and malignant progression (*Qian & Pollard, 2010*). As a result, CCL7 plays an indirect role in RCC through the let-7d-CCL7-macrophage axis. Furthermore, in high-grade metastatic RCC, upregulated expression of CCL7 is particularly prominent, and let-7d is apparently downregulated, which emphasizes the significance of CCL7 in tumor invasion and metastasis (*Su et al., 2014*). In response to mechanical stimulation, PC3 prostate cancer cells secrete more pro-metastatic factors, including CCL7 and TGF-β, and PC3 cells adopt an osteoblast-like phenotype for bone colonization. These events accelerate the growth and bone metastasis of prostate cancer (*Gonzalez et al., 2017*).

### Cancer-associated fibroblasts derived

CCL7 secretion is increased after IL-1β produced by cancer cells activates the NF-κB signaling pathway in cancer-associated fibroblasts (CAFs). CCL7 promotes tumorigenicity, while knockdown of *Ccl7* greatly undermines tumorigenesis. Further study showed that CCL7 mainly promotes breast cancer cell proliferation via binding to its receptor CCR1 (*Rajaram et al., 2013*). IL-1α secreted by oral squamous cell carcinoma (OSCC) induces CCL7 release from activated stromal fibroblasts and stimulates CAFs proliferation. At the same time, CCL7 generated by CAFs is the main promoter of OSCC cell migration and invasion, guides cytoskeletal transformation and provokes membrane ruffling and cell dissemination (*Bae et al., 2014*; *Jung et al., 2010*).

CAFs behave in a collaborative manner with CCL7 to influence tumor migration. Via interacting with tumor cells, activated CAFs enhance the secretion of extracellular matrix modulators to promote tumor migration (*Xu et al., 2015*). Compared with peri-tumor fibroblasts, CAFs are more numerous and express a higher quantity of mesenchymal markers; furthermore, CAFs significantly increase HCC cell migration by inducing EMT in HCC cells *in vitro* (*Fransvea et al., 2008*). CAFs, which have high expression levels of CCL7 and other chemokines, have a powerful effect on HCC metastasis *in vivo* (*Liu et al., 2016*). Additionally, HCC migration induced by CAFs is partially blocked by CCL7 neutralizing antibodies. CCL7 activates the TGF-β pathway by enhancing Smad2 phosphorylation, and blocking the TGF-β pathway markedly inhibits the effects of CCL7 on tumor migration and invasion, highlighting the role of CCL7 in regulating tumor progression by influencing the tumor microenvironment via the TGF-β pathway (*Liu et al., 2016*). Notably, a phase II study on the TGF-β pathway was launched in HCC patients (*Herbertz et al., 2015*). In a co-culture system of CAFs and laryngeal squamous cell carcinoma, CCL7 protein levels were elevated, accompanied by rapid tumor cell

proliferation, but the influence of CCL7 was negligible compared with that of CXCL12 in this study (*Wang et al., 2017*).

### Tumor-associated monocytes/macrophages derived

Tumor-associated macrophages, an important type of immune cell, facilitate tumor angiogenesis and restrain T cell-mediated antitumor reactions, thereby having a positive effect on the tumor development process (*Yang & Zhang, 2017*). In human liver macrophages, also known as Kupffer cells (KCs), increased CCL7 levels create a favorable microenvironment for colorectal cancer liver metastasis (CRLM) (*Mohr et al., 2017*). Alcoholic liver damage is considered a high risk factor for CRLM (*Rasool et al., 2013*). A series of cascades starts from alcohol-stimulated KCs expressing CCL7 and proinflammatory cytokines, among other factors. Then, these paracrine stimuli drive the latent capability of hepatic stellate cells (HSCs), enabling them to undergo a phenotypic change and become an important component of the pro-metastatic liver microenvironment. The activation of HSCs and the accumulation of smooth muscle actin (SMA) often enable the rapid and facile metastasis and recurrence of CRC after surgery (*de la et al., 2001*; *Friedman, 1999*; *Kang, Gores & Shah, 2011*). Additionally, these factors expressed by KCs are known to be involved in matrix degradation, liver parenchymal remodeling, tumor cell adhesion, and colonization (*Mohr et al., 2017*).

Patients with pancreatic ductal adenocarcinoma (PDAC), which is associated with T helper 2 (Th2) lymphoid cell infiltration in the pathology analysis, often have poor clinical outcomes (*Protti & De Monte, 2012*). Thymic stromal lymphopoietin (TSLP), produced by activated CAFs, is capable of activating myeloid DCs to acquire Th2-polarizing capability in the tumor microenvironment (*Tjota & Sperling, 2014*). Remodeled Th2 cells migrate to tumor-draining lymph nodes (TDLNs) and secrete more IL-3. Secreted IL-3 activates basophils that express IL-4 *in vivo*. In TDLNs, CCL7 mRNA levels are markedly increased after stimulation of monocytes with TSLP. CCL7 secretion from monocytes significantly contributes to the recruitment of basophils into TDLNs. CCL7 antibodies partially block the migration of basophils and CAFs *in vitro*. IL-4-positive basophils show greater accumulation in TDLNs than in non-TDLNs, which is relevant to Th2 inflammatory responses and indicates a poor prognosis in patients with a high proportion of basophils (*De Monte et al., 2016*). CCL7, which is irregularly released by immature monocytic myeloid cells (IMMCs) or neutrophils, somewhat increases the migration and invasion of human non-small cell lung cancer cells, suggesting the possibility that CCL7 exerts its carcinogenesis properties as a chemoattractant for neutrophils involved in the formation of the tumor microenvironment (*Durrans et al., 2015*; *Michalec et al., 2002*).

### Cancer-associated adipocyte-derived

Adipose tissues are generally recognized as an active endocrine organ, secreting hormones, growth factors, chemokines, and proinflammatory molecules (*Ouchi et al., 2011*) that advance the evolution of diverse diseases and certain types of cancer (*Hursting & Dunlap, 2012*). After communicating with cancer cells, adipocytes are transformed into
cancer-associated adipocytes (CAAs), which adopt a fibroblast-like phenotype and secrete a large number of bioactive factors, thereby promoting tumor proliferation and metastasis (*Bochet et al., 2013*; *Nieman et al., 2013*). Epidemiological studies show that obese patients are more susceptible to prostate carcinoma, breast cancer and CRC and often have poor outcomes (*Allott, Masko & Freedland, 2013*; *Parker et al., 2013*). CCL7 is found both in preadipocytes and adipocytes in a 3T3-L1 cell differentiation model (*Kabir, Lee & Son, 2014*).

The level of CCL7, which is generated by CAAs, increases gradually from the peripheral prostatic gland to the periprostatic adipose tissue. This expression pattern favors the migration of CCR3-positive prostate cancer cells along a chemokine gradient (*Laurent et al., 2016*). The establishment of a chemokine axis is the basis for the directional migration of cancer cells. Adipose tissues dissected from obese mice release more CCL7, and higher CCL7 expression promotes prostate cancer cell migration. Moreover, depletion of the CCL7/CCR3 axis completely abolishes the ability of obesity to promote tumor metastasis, which reflects the importance of the CCL7/CCR3 axis in prostate cancer in the context of obesity (*Laurent et al., 2016*). Intriguingly, compared with lean controls, those with diet-induced obesity show increased sensitivity to *Helicobacter felis*-associated proinflammatory signals. More adipose inflammatory factors can be released by CAAs in gastric cancer, indirectly increasing CCL7 production. The increase in CCL7 positively amplifies the proinflammatory reaction feedback loop, modulating IMMC mobilization in gastric tissues and the T helper 17 (TH17) response, which are conducive to the formation of the gastric tumor microenvironment (*Ericksen et al., 2014*).

### Other cells derived

Upon binding to its receptor CCR2, CCL7 plays a pivotal role in the recruitment of macrophages by tumor cell-derived exosome-educated MSCs (*Lin et al., 2016*). MSCs separated from spontaneous mouse lymphomas (L-MSCs) also express higher levels of CCR2 ligands (CCL7 and CCL2) than BM-MSCs do (*Ren et al., 2012*). Tumor-resident MSCs overexpressing CCL7 are liable to provide a suitable microenvironment for tumor growth by increasing the interaction with surrounding immune cells and by promoting macrophage infiltration. Additionally, abundant macrophages around the periphery of the tumor promote tumorigenesis and increase malignant behavior (*Qian & Pollard, 2010*). Astrocytes secrete more CCL7 upon stimulation with cyclooxygenase 2 and prostaglandins. Expression of a critical stem cell gene, Nanog, is increased in response to CCL7 overexpression and promotes the self-renewal of initiating breast carcinoma cells that are transferred to brain tissues (*Wu et al., 2015*).

CCL7 produced by matrix cells plays an important role in the BM homing of human multiple myeloma (MM) cells. CCR2, the receptor for CCL7, is expressed in human metastatic MM cell lines (HMCLs; Karpas, LP-1, and MM5.1) and in primary MM cells. Moreover, CCL7, CCL2, and CCL8 are expressed in stromal cells of the BM. Exogenous CCL7 mediates the concentration-dependent stimulation of MM cell migration. A blocking CCR2 monoclonal antibody and/or neutralizing CCL7 antibodies abolish some of the pro-migration effects of conditioned medium from BM stromal cells on MM

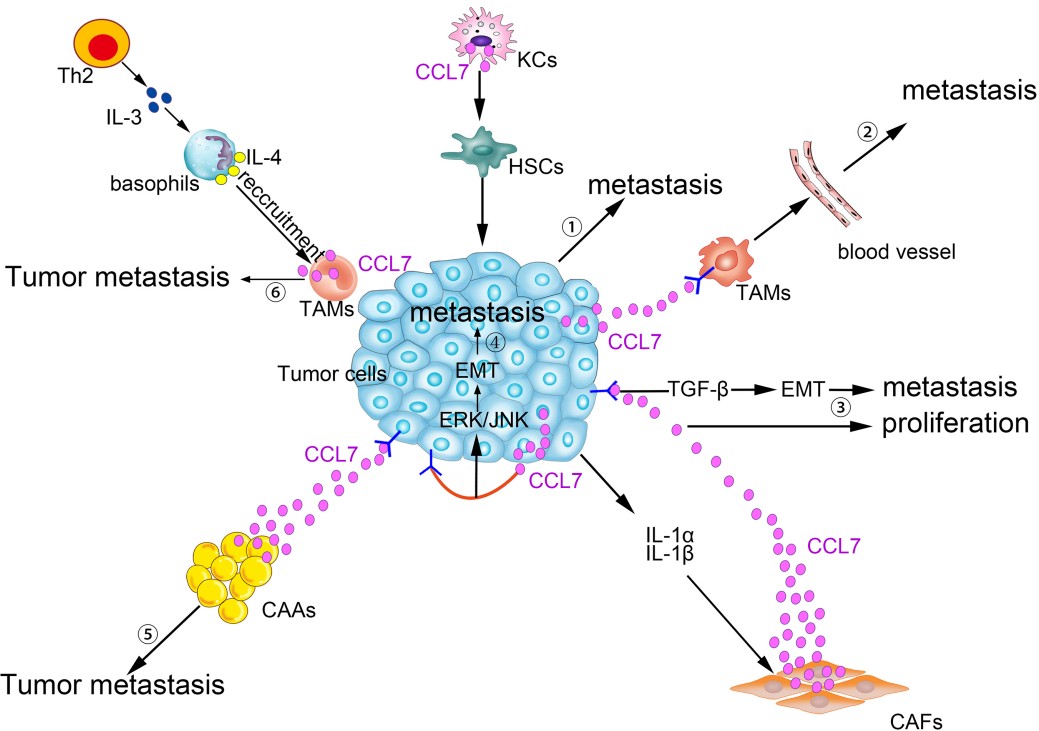

**Figure 2 Pro-tumor effects of CCL7 in tumor microenvironment.** ① Alcohol-stimulated KCs express CCL7 to facilitate HSCs phenotypic transform, consequently, CRC liver metastases rate increases. ② Intratumor overexpression of CCL7 can enhance vascular permeability by recruiting TAMs to promote RCC brain metastasis. ③ The level of CCL7 is increased after being stimulated by IL-1α and IL-1β in CAFs. Excessive CCL7 promotes EMT progression via the TGF-β pathway, resulting in increased tumor invasion and metastasis and CCL7 also promotes tumor proliferation. ④ CCL7 derived from tumor cells binds to CCR3, and then participates in the EMT process and promotes tumor metastasis via ERK/JNK signal pathway. ⑤ Interacting with its receptors, CCL7 secreted by CAAs can instruct the tumor cells along the concentration gradient of CCL7 dissemination. ⑥ TAMs-secreted CCL7 promotes the recruitment of basophils into TDLNs, which facilitates tumor metastasis.

(*Vande Broek et al., 2003*). CCL7 secreted by matrix cells is also involved in the bone metastasis of breast cancer cells. In a breast cancer cell and osteoblast co-culture system, MMP-13 stimulates the matrix to produce more CCL7. On the one hand, CCL7 itself participates in the recruitment of monocytes and osteoblasts (*Yu et al., 2004*), exerting its pleiotropic tumorigenic roles in breast cancer homing to bone and in metastatic growth (*Bar-Shavit, 2007*). On the other hand, CCL7 is cleaved by MMP-13, and truncated CCL7 abrogates the action of its corresponding receptors. Targeted cleavage of CCL7 by MMP-13 is a part of a negative feedback loop, which in turn increases the secretion of MMP-13 and osteolysis. Then, malignant MDA-MB-231 cells are liable to migrate to the bone (*Morrison et al., 2011*). Generally, as shown in Fig. 2, CCL7 plays a crucial role in the crosstalk between tumor cells and stromal cells to promote tumorigenesis.

## Antitumor effects of CCL7

CCL7 is a well-characterized chemokine that is generally acknowledged as an inflammatory cytokine. Its inflammatory activity depends primarily on its ability to

attract diverse leukocyte subgroups. Nonetheless, T lymphocytes and DCs activated by CCL7 play an important role in mobilizing the immune response to resist tumor growth, and CCL7 is expected to be an appealing antitumor molecular target under certain circumstances. Stably overexpressed CCL7 in tumor cells can restrain tumor growth in multiple mouse tumor models, including models of human cervical tumor (*Wetzel et al., 2001*), melanoma (*Wetzel et al., 2007*), malignant glioblastoma (*Geletneky et al., 2010*), PDAC (*Dempe et al., 2012*), and CRC (*Hu et al., 2002*).

A particular exogenous model is necessary to successfully achieve this function; this model utilizes the properties of the autonomous parvovirus minute virus of mice (MVMp) or the rodent parvovirus H-1PV. As small nuclear-replicating DNA viruses, autonomous parvoviruses replicate independently of a helper virus (*Cotmore & Tattersall, 1987*). The replication of genus parvovirus in proliferating cells occurs in a dominant manner, and these viruses have onco-suppressive activity in laboratory animals via their lytic life cycle. The preferential replication of this virus in cancer cells makes it possible to target recombinant MVMp/CCL7 parvoviruses to cancer cells. Then, hCCL7 can be overexpressed by tumor-bearing animals after subcutaneously implantation of mouse mastocytoma cells harboring recombinant virus (*Haag et al., 2000*; *Wetzel et al., 2001*). CCL7 overexpression increases the recruitment of leukocytes and triggers type I T cell-dependent reactions, evoking an antitumor cascade (*Fioretti et al., 1998*).

hCCL7 restricts the progression of tumor growth in a concentration-dependent manner. At doses up to 10 RU/cell, no B78/H1 melanoma cells could be successfully implanted in mice in the presence of MVMp/CCL7. In K1735 melanoma, tumor formation and growth were dramatically reduced (*Wetzel et al., 2007*). The different extents of inhibition may depend on the intrinsic characteristics of the tumor itself. Notably, neither the wild-type viral vector group nor the empty vector group showed strong antitumor activity, although the wild-type virus replicates completely in permissive cancer cells *in vitro* (*Dempe et al., 2010*; *Wetzel et al., 2007*). This result may be explained by the absorption of tumor cell-generated virus by other cells in the microenvironment (*Wrzesinski et al., 2003*). Intratumoral overexpression of CCL7 promotes the recruitment of additional immune cells and triggers T helper 1 (Th1) responses. Furthermore, the elimination of the antitumor effect by complete depletion of CD4$^+$ T cells, CD8$^+$ T cells or NK cells indicates that antineoplastic activities require the involvement of T lymphocytes and NK cells (*Wetzel et al., 2007*). Only NK cells, not DCs, are involved in xenograft models of human PDAC (*Dempe et al., 2012*). As cells that respond downstream of CCL7, both T lymphocytes and NK cells can turn on a death pathway that is characterized by the production of granzymes and perforin (*Russell & Ley, 2002*). The release of IFN-γ in the tumor is a marker of a Th1 response, which is a key for the antitumor effect of CCL7 and amplifies the cytocidal activities of leukocytes by cooperating with the perforin/granzyme-mediated system. CCL7 not only inhibits tumor growth but also completely blocks tumor metastasis in a mouse colon cancer CMT93 model (*Hu et al., 2002*). Interestingly, greater antineoplastic activity is induced in MHC I-negative carcinoma by increasing the sensitivity of tumor cells to immune cell-mediated

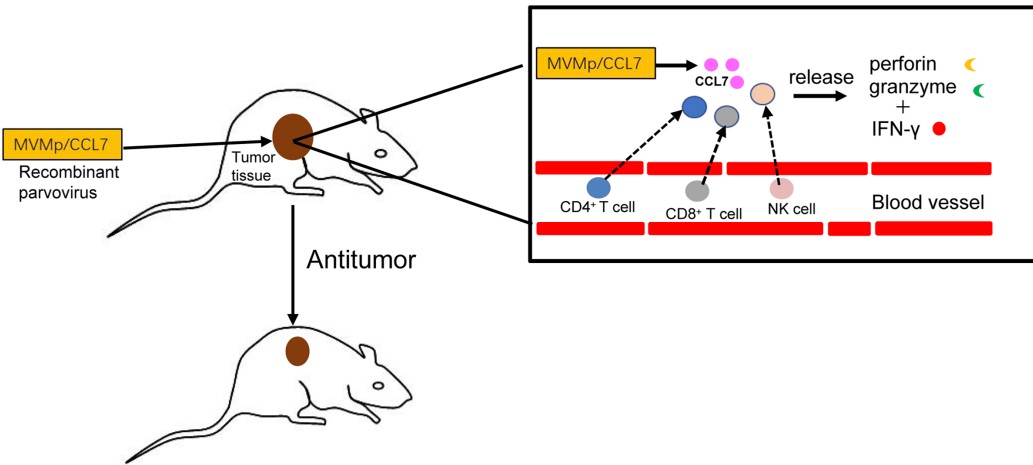

**Figure 3 Antitumor effects of CCL7.** The recombinant MVMp/CCL7 parvovirus is injected into tumor-bearing mice, and then numerous CCL7 is released from experimental animals. Overexpressed CCL7 exerts antitumor effect in a dose-dependent manner by recruiting CD4+, CD8+T lymphocytes and NK cells around the tumor. Recruited T lymphocytes and NK cells produce large amounts of perforin and granzymes, which cooperate with IFN-γ to amplify the cytocidal activities. Therefore, a large number of cancer cells is lysed and died.

death pathways in both human PDAC cells and mouse melanoma B78/H1 cells (*Angelova et al., 2009*; *Wetzel et al., 2007*).

In conclusion, as shown in Fig. 3, the delivery of CCL7 to tumor cells by parvovirus vectors can decrease malignant behavior and inhibit tumor growth. Thus, chemokine CCL7 may act as a strong attractant for immunocompetent cells to strengthen the immune response, and it might be expected to serve as important adjuvants in cancer vaccines or considered for the treatment of certain human cancers by direct intratumoral application in the near future.

## CCL7 and clinical prognosis

Abnormally elevated expression of CCL7 often predicts a poor prognosis. In T3 stage RCC tissues, CCL7 levels negatively correlate with let-7d expression. Low let-7d expression levels and high CCL7 levels predict advanced T stage and high tumor grade (*Su et al., 2014*). Incredibly, human breast carcinoma and diffuse-type gastric cancer also show this correlation (*Wu et al., 2006*). CCL7 expression is higher and more pronounced in the cytoplasm of gastric cancer cells than matched adjacent non-neoplastic tissues. Kaplan–Meier survival curves for disease-free survival illustrate that patients with higher CCL7 expression have shorter survival than those with lower CCL7 expression. Increased CCL7 expression indicates extensive invasion, lymph node metastasis, and higher tumor-node-metastasis stage (*Hwang et al., 2012*). In patients of CRC, remarkable CCL7 expression often means liver metastases and also represents poor prognosis (*Cho et al., 2012*).

## CONCLUDING REMARKS AND PERSPECTIVES

CCL7 is secreted by many types of cells, including stromal cells and tumor cells. As a member of the CC chemokine subgroup, CCL7 mainly exerts chemotactic activity,

attracting monocytes, DCs, and activated T lymphocytes, among others, to invasion sites, which is a prerequisite for mediating inflammatory reactions and exerting antiviral effects. A growing body of evidence indicates that excessive CCL7 expression is also favorable in terms of the development and progression of some types of cancer, including breast cancer, RCC, and CRC. In contrast, CCL7 has an inhibitory effect on tumor progression by utilizing the characteristics of chemotactic immune cells. Although there has been much exhaustive research on CCL7, some problems remain to be addressed. First, the previously reported signaling pathways mainly relate to the promotion of tumor metastasis, and the correlation between CCL7 and tumor proliferation is unclear. CCL7 has 71 percent sequence homology with CCL2, but it is unknown whether CCL7 mimics CCL2-mediated tumor angiogenesis, which is favorable for tumor proliferation. Second, the mechanisms underlying the tumor-promoting effects of CCL7 have not been completely elucidated. The identification of new associated signaling pathways will undoubtedly lead to more efficient and successful clinical applications in the future. Additionally, delayed tumor growth can be achieved upon neutralization of CCL7 by using diverse therapeutic methods, like monoclonal antibodies, gene silencing. Given the high chemokine synthesis rates, the use of monoclonal antibodies requires improving conventional bioanalytical tools to measure the free ligand *in vivo* to determine the amount of targeted drug and evaluate the efficacy. Designed to increase the efficiency of gene silencing, a new gene silencing approach would be adopted that involves complexing siRNAs to TAT peptides (transactivated transcript peptides derived from HIV-1) through non-covalent calcium cross-linking. However, any treatment that attempts to reduce the expression of its receptors for therapeutic purposes requires careful consideration because a receptor can bind multiple ligands; blockade of a chemokine receptor may inhibit other ligands involved in the signaling pathway in most instances. And generating specific antibodies against seven-transmembrane receptors is also very challenging. Finally, vaccination based on the antitumor properties if CCL7 must be evaluated in extensive preclinical studies. Therefore, more detailed knowledge of the effect of CCL7 on tumorigenesis remains to be obtained.

## ACKNOWLEDGEMENTS

We thank that Pro. Daya Luo made valuable suggestions for us in this manuscript. Special thanks to Pro. Xiaohua Yan for always support our work in several aspects.

### Funding

This work was supported by the National Natural Science Foundation of China (No. 81760509). The funders had no role in study design, data collection and analysis, decision to publish, or preparation of the manuscript.

### Grant Disclosures

The following grant information was disclosed by the authors:
National Natural Science Foundation of China: No. 81760509.

## Competing Interests

The authors declare that they have no competing interests.

## Author Contributions

- Yangyang Liu prepared figures and/or tables, authored or reviewed drafts of the paper.
- Yadi Cai prepared figures and/or tables, authored or reviewed drafts of the paper.
- Li Liu contributed reagents/materials/analysis tools.
- Yudong Wu contributed reagents/materials/analysis tools.
- Xiangyang Xiong approved the final draft, reviewed drafts of the paper.

## Data Availability

This article did not generate any data or code, as it is a literature review.

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
