# Peer review of "Crucial biological functions of CCL7 in cancer"

_PeerJ, doi:10.7717/peerj.4928_

## Round 0.1 · original submission · Minor Revisions

Your article has been found to be timely and of great interest. The reviewers requested minor revisions and if you can address these concerns, we will be pleased to reconsider your submission.

Reviewer 1 ·

Basic reporting

This is an attractive and well written review. The figures are well designed and the article covers an important field. However, the survey methodology appears to have left certain key elements in need of additional discussion. The information on the expression of key CCL7 receptors (CCR1, -2, -3) on tumor cells and CCL-7 mediated signaling responses mediated within the tumor cells is too sparse. Brief mention is given to CCR1 expression on breast cancer cells. However much more recent information is available on CCR1 expression not only by breast cancer (Cao Faseb J 2017, Shin Oncotarget 2017), but colorectal cancer (PMID 26383527), and lung cancer (PMID 18972130) for instance. Another area that is lacking involves therapeutic options: How do the authors propose targeting CCL7 and which methodologies hold more promise (e.g. peptide, shRNA, antisense oligo, receptor aptamer etc)? Do the authors believe that, and how should CCL7 antagonists be developed for cancer treatment?

Experimental design

As previously mentioned, the survey methodology appears to have left certain key elements in need of additional discussion.

Validity of the findings

No comment

Additional comments

No comment

Reviewer 2 ·

Basic reporting

no comment

Experimental design

no comment

Validity of the findings

no comment

Additional comments

This is a comprehensive review that summarized both pro-tumor and anti-tumor roles of CCL7 derived from a variety of tumor and tumor-associated cells. While the authors aimed to review all available literatures on this topic, it would be important to connect every piece of observation in a logic way. Some issues need to be addressed.

Specific comments:
1. The gene symbol for human CCL7 should be capitalized (CCL7).
2. Line 45: “the first to be discovered” should be “the first have been discovered”.
3. Line 56: “and under pathological conditions (tumor cells)” should be “and in tumor cells under pathological conditions”.
4. Line 57: Please specify “immune-related cells” as all examples are leukocytes.
5. Line 60-63: Please explain briefly how CCL7 is related to these diseases.
6. Line 94-105: This paragraph is to describe the regulation of CCL7 expression. The first sentence “Chemokines often have shared and exclusive chemokine receptors.” is actually irrelevant and it may be moved to the paragraph of “CCL7 RECEPTORS”. In addition, the content of the text here and that of Figure 1 are completely different. How is CCL7 induced or suppressed by cytokines/stimuli? Which responsive elements in its promoter/enhancer are involved? Which cells are examined? All of these information together with references should be included in both the text and Figure 1 consistently.
7. Line 98-100: Please revise this sentence “The upregulation of CCL7 is associated with IL-1β, IFN-α, IFN-β, and measles virus in peripheral blood mononuclear cells (PBMC) (Menten et al. 1999).” to clarify its meaning.
8. Line 108: “subpopulation” should be “subtype”.
9. Line 141: “CCL7 also may affects” should be “CCL7 may also affect”.
10. Line 161: Please revise this sentence “CCL7 greatly impacts both antiviral and anti-bacterial immune responses.” as the role of CCL7 in antifungal immunity is discussed later in the same paragraph.
11. Line 166: Please describe “Tip-DC” briefly.
12. Line 176-186: The anti-tumor role of CCL7 should be mentioned in this paragraph.
13. Line 203: “the ERK-JNK signaling pathway” is parallel. Both ERK and JNK are MAPK.
14. Line 206-216: The regulation of CCL7 by microRNA Let-7d should be discussed in the paragraph of “THE STRUCTURE AND REGULATION OF CCL7”.
15. Line 219: “TGBβ1” should be “TGF-β1”.
16. Line 223: What is the relationship between CCL7 secretion and IL-1β?
17. Line 306: “do BM-MSCs” should be “BM-MSCs do”.
18. The paragraph of “CCL7 and clinical prognosis” should be moved after the paragraph of “Antitumor effects of CCL7”.
19. In Figure 2, the MAPK signaling pathway should appear inside tumor cells since the other arrows around indicate different tumor-associated cells. Please also correct the MAPK-ERK-JNK pathway in this figure.
20. In Figure 3, please clarify the anti-tumor role of CCL7 in detail.

Reviewer 3 ·

Basic reporting

Please find attached PDF file.

Experimental design

Please find attached PDF file.

Validity of the findings

Please find attached PDF file.

Annotated reviews are not available for download in order to protect the identity of reviewers who chose to remain anonymous.

---

## Round 0.2 · accepted · Accept

I confirm that your revisions are acceptable and the manuscript is now suitable for publication. Please be sure to check your English grammar.

#